# Seasonal Changes in Sleep Patterns in Two Saskatchewan First Nation Communities

Chandima P. Karunanayake [1,*], Vivian R. Ramsden [2], Clifford Bird [3], Jeremy Seeseequasis [4], Kathleen McMullin [1], Mark Fenton [5], Robert Skomro [5], Shelley Kirychuk [1,5], Donna C. Rennie [6], Brooke P. Russell [1], Niels Koehncke [1,5], Thomas Smith-Windsor [7], Malcolm King [8], Sylvia Abonyi [8], James A. Dosman [1,5] and Punam Pahwa [1,8]

[1] Canadian Centre for Health and Safety in Agriculture, University of Saskatchewan, 104 Clinic Place, Saskatoon, SK S7N 2Z4, Canada; kathleen.mcmullin@usask.ca (K.M.); shelley.kirychuk@usask.ca (S.K.); bpr053@mail.usask.ca (B.P.R.); niels.koehncke@usask.ca (N.K.); james.dosman@usask.ca (J.A.D.); pup165@mail.usask.ca (P.P.)
[2] Department of Academic Family Medicine, University of Saskatchewan, West Winds Primary Health Centre, 3311 Fairlight Drive, Saskatoon, SK S7M 3Y5, Canada; viv.ramsden@usask.ca
[3] Community B, P.O. Box 250, Montreal Lake, SK S0J 1Y0, Canada; c.bird@sasktel.net
[4] Community A, P.O. Box 96, Duck Lake, SK S0K 1J0, Canada; jccquasis@willowcreehealth.com
[5] Department of Medicine, University of Saskatchewan, Royal University Hospital, 103 Hospital Drive, Saskatoon, SK S7N 0W8, Canada; mef132@mail.usask.ca (M.F.); r.skomro@usask.ca (R.S.)
[6] College of Nursing, University of Saskatchewan, 104 Clinic Place, Saskatoon, SK S7N 2Z4, Canada; donna.rennie@usask.ca
[7] Victoria Hospital, Prince Albert, SK S6V 4N9, Canada; dr.tom@sasktel.net
[8] Department of Community Health and Epidemiology, College of Medicine, University of Saskatchewan, 107 Wiggins Road, Saskatoon, SK S7N 5E5, Canada; malcolm.king@usask.ca (M.K.); sya277@mail.usask.ca (S.A.)
* Correspondence: cpk646@mail.usask.ca; Tel.: +1-306-966-1647

**Abstract:** Sleep is crucial for maintaining the recovery and restoration of the body and brain. Less sleep is associated with poor mental and physical performance. Seasonal changes in sleep patterns can be observed. This paper examines seasonal effects on sleep timing, duration, and problems in two Cree First Nation communities in Saskatchewan, Canada. Data were available from a community survey of 588 adults aged 18 years and older (range: 18–78 years) with 44.2% males and 55.8% females. Results are presented using descriptive statistics and a binary logistic-regression model to identify the association between seasonal changes in sleep patterns, and demographic, social, and environmental factors. The participants reported sleeping the least during the spring and summer months and sleeping the most during the fall and winter months. This was further confirmed by sleep hours and the lower proportion of recommended hours of sleep during the spring and summer, and a higher proportion of longer sleep duration during the fall and winter months. There was no significant variation in sleeping onset and wake-up times by season. Overall, there were no significant differences in the prevalence of sleep deprivation, insomnia, and excessive daytime sleepiness by season. When stratified by age group and sex, some differences existed in the prevalence of sleep problems by season. More than two-thirds (68.6%) of the participants reported that there was a change in sleep patterns across seasons, and about 26.0% reported a very or extremely marked change in sleep patterns across seasons. Changes in sleep patterns by season were related to money left at the end of the month and damage caused by dampness in the house.

**Keywords:** seasonal changes; sleep patterns; First Nations; adults

## 1. Introduction

Studies showed that seasonal changes are associated with sleep duration and problems [1–4]. Sleep onset and offset are also reported to vary by the length of daylight

hours [5,6]. The geographical coordinates of Saskatchewan, Canada are (latitude: 55° 0′ 0.0000″ N, longitude: 106° 0′ 0.0000″ W) [7]. Figure 1 shows the hours of daylight illumination during 2018 in Saskatchewan [8]. The province of Saskatchewan experiences wide seasonal variations in the length of daylight hours. The maximal length of daylight hours was 17.6 h in month of June, and the minimal length of daylight hours was 6.97 h near the end of December. Saskatchewan receives an average of 2206 h of sunshine per year, more sunshine than any other province in Canada [9].

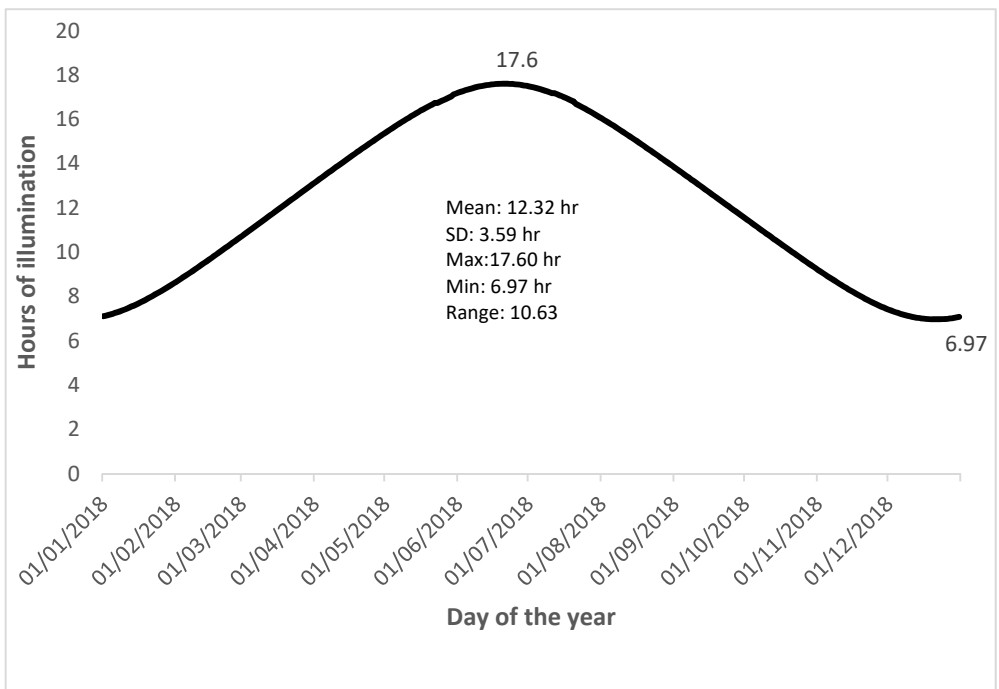

**Figure 1.** Hours of daylight illumination during 2018 in Saskatchewan.

Symptoms of depression and mood disorders are reported to vary with seasons [10–13]. Lukmanjini et al. [14] found a high proportion of sleep problems among Canadian adults (25+ years) during fall and winter months (21 September to 21 March). Hardin et al. [1] reported seasonal variation of sleep duration among the general population of the United States, with participants sleeping more in winter. No studies explored seasonal changes in sleep time and duration in First Nations people in Canada. First Nations comprise one of the three groups of Indigenous peoples who are the descendants of the original inhabitants of North America (the other two being Inuit and Métis). First Nations peoples have unique cultures, languages, and ceremonies [15,16]. All Indigenous peoples in Canada were impacted by historical perspectives and colonization, which produced inequities in social and structural determinants of health [17]. This paper examines the seasonal effects on sleep time, duration, and problems in two Cree First Nation communities in Saskatchewan, Canada, and identifies the association between seasonal changes in sleep patterns and demographic, social, and environmental factors. The expected outcomes of this paper help to identify the seasonal effects on sleep and the factors related to seasonal changes in sleep patterns.

## 2. Methods

### 2.1. Study Sample

The data for this study came from a baseline survey undertaken during the First Nations Sleep Health Project (FNSHP) conducted in partnership with two Cree First Nation communities (Communities A and B) in Saskatchewan in 2018–2019. The purpose of the FNSHP was to examine the relationships between sleep disorders, and risk factors and

comorbidities, and to evaluate local diagnosis and treatment. The study was approved by the University of Saskatchewan's Biomedical Research Ethics Board (certificate no. Bio #18-110) and adhered to Chapter 9 (Research Involving the First Nations, Inuit, and Metis Peoples of Canada) in the Tri-Council Policy Statement: Ethical Conduct for Research Involving Humans [18]. Community-level consent was secured through a long-term collaboration between the University of Saskatchewan and the research team members in both communities. Individual participants provided written consent following an informed-consent process.

### 2.2. Data Collection

Trained research assistants from each community conducted the baseline surveys in their respective communities. Adults 18 years and older were invited to the community health and youth centre to complete the interviewer-administered questionnaires and clinical assessments. A pamphlet describing the study and an invitation to participate were distributed by the research assistants during local community events such as Treaty Days and during door-to-door canvassing. Simultaneously, there was a social-media campaign to invite the community members to participate in the survey. The survey collected information on demographic variables, individual and contextual determinants of sleep health, self-reported height and weight, and objective clinical measurements. This manuscript is based on data from the questionnaires. Demographic information about participants including age, sex, body-mass index, education level and money left at the end of the month, lifestyle factors, house environment, medical history, and sleep-health information was obtained from the survey questionnaire.

### 2.3. Variables

The following questions addressed seasonal effects on sleep patterns using the modified Seasonal Pattern Assessment Questionnaire (SPAQ) [19]: (i) "At what time of year do you sleep least?"; (ii) "at what time of year do you sleep most?"; (iii) "how much does your sleep change with the seasons?"; (iv) "approximately how many hours of each 24 h day do you sleep each season?" Three other questions were asked about usual bedtime at night, usual time of wake up in the morning, and time to fall asleep at night: "During the past month, what time have you usually gone to bed?", "during the past month, what time have you usually gotten up in the morning?", and "during the past month, how long (in minutes) has it taken you to fall asleep each night?" In addition to the relationships between season (based on astronomical start or end dates of a season and considering self-reported date to correspond to the season), data were reported and sleep problems such as excessive daytime sleepiness (EDS), sleep deprivation, and insomnia were assessed. Sleep duration was calculated by subtracting time to fall asleep from time in bed during the night. Sleep deprivation was defined as sleep duration of less than 7 h per night (less than the recommended optimal sleep duration per night [20]). Clinical insomnia was defined using the Insomnia Severity Index (ISI) score, which indicated to be equal or greater than 15 [21–23]. Excessive daytime sleepiness (EDS) was defined as having occurred over the past month using the Epworth Sleepiness Scale (ESS) at a value >10 [24,25].

### 2.4. Statistical Analysis

Statistical analyses were conducted using SPSS version 27 [26] and R software [27]. Descriptive statistics, mean, median, and standard deviation (SD) are reported for continuous variables. For categorical variables, frequency and percentages (%) are reported. Chi-squared tests were used to determine the bivariable association of insomnia and sleep-deprivation prevalence by season. When expected cell frequency was <5, Fisher's exact test *p* values were reported using R function fisher.test, which used 2000 simulations. Sleep duration hours by season using median test and pairwise comparisons were compared using Mood's median test [28,29]. Two sample-proportion tests were conducted using the Z test [30]. Sleep change by season (very or extremely marked, or no-to-moderate

change) was further analysed using a binary logistic-regression model to determine the association between demographic, social, and environmental factors [31]. These cut-offs for sleep change were decided on the probable significant change in sleep habits with very or extremely marked change. On the basis of bivariable analysis, variables with $p < 0.20$ were considered for the multivariate model. All variables that were statistically significant ($p < 0.05$), and important clinical factors (sex, age) were retained in the final multivariable model. Interactions between potential effect modifiers were examined and retained in the final model if the $p$ value was <0.05. The strengths of the associations were presented by odds ratios (OR) and their 95% confidence intervals (CI) [31].

## 3. Results

A total of 588 participants completed the baseline survey: 418 individuals from Community A and 170 individuals from Community B. Demographic information about the participants was presented elsewhere and is only briefly described here [32,33]. Mean age $\pm$ SD of the 588 study participants was $40.0 \pm 15.3$ years, and the age was in the range of 18–78 years. There were 44.2% males and 55.8% females in this study. About 65% of the participants were identified as being overweight or obese. Duration of sleep was available for 96% (567/588) of the adults who participated. The mean reported duration of sleep was $8.18 \pm 2.28$ h, and the median sleep duration was 8.00 h. The prevalence of sleep deprivation was 25.4% (144/567) [32,33].

Of the participants, 77% (455/588) responded to questions related to time of the year during which they slept least and most (Figure 2). According to the time of the year by months, participants reported that they slept the least during the months of May–August (spring and summer), and they slept the most during the months of November–February (fall and winter).

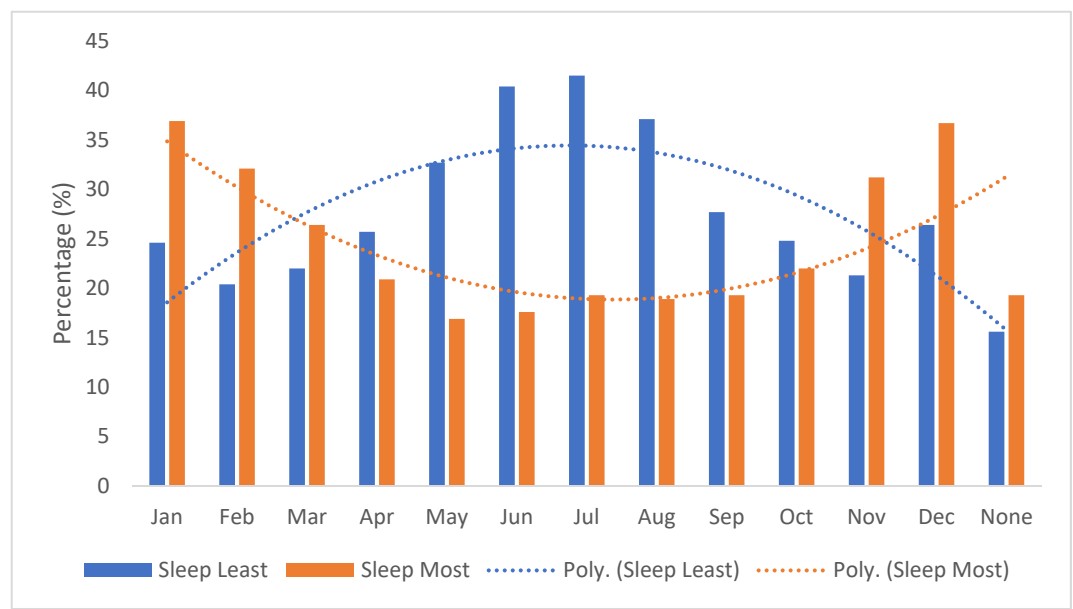

**Figure 2.** Percentage of time of the year of sleeping least and most ($n = 455$).

Figure 3 shows the hours of sleep during the four seasons. A lower proportion of the recommended hours of sleep (7–9 h) [20] occurred during summer (45.2% vs. fall, 52.9%; spring, 50.7%; and winter, 49.9%) and a higher proportion of sleep of longer duration (>9 h) occurred during the winter (15.6% vs. spring, 6.4%; summer, 7.2%; and fall, 9.2%) (Figure 3).

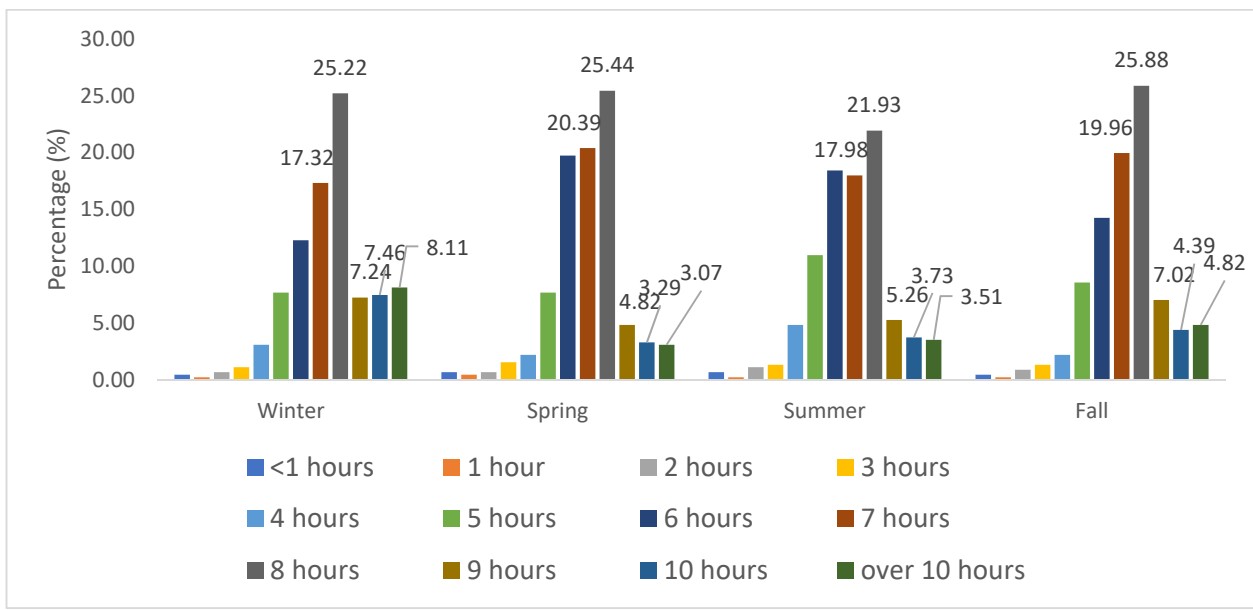

**Figure 3.** Hours of sleep during the four seasons.

Most participants reported their usual bedtime as between 12:00 and 2:00 A.M. However, there was no significant variation between seasons. Usual wake-up time for most was 6:00–8:00 A.M., and there was a considerable proportion of people who woke up later in the morning, at 8:01–10:00 A.M. (Table 1). Even though we did not see any difference of usual bedtime by seasons, we saw a significant difference in wake-up times ($p = 0.014$), with a smaller proportion of respondents who woke up early in winter compared to the three other seasons (Table 1). The median sleep duration in hours varied by season, and there was a significant difference between sleep duration in spring and summer. There were no differences in fall or winter (Table 2).

**Table 1.** Sleeping and wake-up times by seasons ($n = 567$).

| Time | Frequency *n* (%) | Seasons *n* (%) | | | |
|---|---|---|---|---|---|
| | | Summer (*n* = 136) | Fall (*n* = 213) | Winter (*n* = 38) | Spring (*n* = 180) |
| **Usual Bed Times (*p* = 0.153)** | | | | | |
| 5:00 PM–8:00 PM | 14 (2.5) | 2 (1.5) | 7 (3.3) | 2 (5.3) | 3 (1.7) |
| 8:01 PM–10:00 PM | 139 (24.5) | 32 (23.5) | 68 (31.9) | 6 (15.8) | 33 (18.3) |
| 10:01 PM–12:00 AM | 176 (31.0) | 43 (31.6) | 61 (28.6) | 15 (39.5) | 57 (31.7) |
| 12:00 AM–2:00 AM | 184 (32.5) | 45 (33.1) | 58 (27.2) | 13 (34.2) | 68 (37.8) |
| 2:01 AM–4:00 AM | 36 (6.3) | 8 (5.9) | 15 (7.0) | 2 (5.3) | 11 (6.1) |
| Other random times | 18 (3.2) | 6 (4.4) | 4 (1.9) | 0 (0.0) | 8 (4.4) |
| **Usual Wake-Up Times (*p* = 0.014)** | | | | | |
| 4:00 AM–6:00 AM | 104 (18.3) | 22 (16.2) | 47 (22.1) | 4 (10.5) | 31 (17.2) |
| 6:01 AM–7:00 AM | 142 (25.0) | 22 (16.2) | 58 (27.2) | 11 (28.9) | 51 (28.3) |
| 7:01 AM–8:00 AM | 136 (24.0) | 31 (22.8) | 49 (23.0) | 8 (21.1) | 48 (26.7) |
| 8:01 AM–10:00 AM | 116 (20.5) | 35 (25.7) | 41 (19.2) | 8 (21.1) | 32 (17.8) |
| 10:01 AM–12:00 PM | 48 (8.5) | 18 (13.2) | 17 (8.0) | 4 (10.5) | 9 (5.0) |
| Other random times | 21 (3.7) | 8 (5.9) | 1 (0.5) | 3 (7.9) | 9 (5.0) |

**Table 2.** Sleeping hours by seasons.

| Seasons ($n$ = 567) | Mean ± SD Hours | Median Hours of Sleep | $p$ Value |
|---|---|---|---|
| Summer ($n$ = 136) | 8.44 ± 0.21 | 8.50 | $H_0$: medians hours of sleep among |
| Fall ($n$ = 213) | 8.26 ± 0.14 | 8.00 | seasons are the same. |
| Winter ($n$ = 38) | 8.28 ± 0.31 | 8.38 | $p$ = 0.048. Since $p$ < 0.05, reject $H_0$ at 5% |
| Spring ($n$ = 180) | 7.87 ± 0.19 | 7.83 | significance level |
| Pairwise comparisons of seasons | | | |
| Seasons | Mood's Median Test Statistic | $p$ Value | Adjusted $p$ value for multiple tests after applying Bonferroni correction |
| Spring–fall | 1.525 | 0.217 | 1.000 |
| Spring–winter | 3.187 | 0.074 | 0.445 |
| Spring–summer | 7.632 | 0.006 | 0.034 |
| Fall–winter | 0.291 | 0.590 | 1.000 |
| Fall—summer | 1.849 | 0.174 | 1.000 |
| Winter–summer | 0.111 | 0.739 | 1.000 |

Overall, there were no significant associations between season and sleep problems (sleep deprivation ($p$ = 0.091), clinical insomnia ($p$ = 0.370), or excessive daytime sleepiness ($p$ = 0.148)). However, there was a higher prevalence of sleep deprivation (40.3%), insomnia (38.5%), and excessive daytime sleepiness (40.4%) in spring when compared to the three other seasons (Table 3). This was stratified by age group and sex.

**Table 3.** Sleep problems by seasons based on astronomical start dates by age group and sex.

| Age Group/Sex | Sleep Problem Yes/No | Seasons $n$ (%) | | | | $X^2$, df = 3 | Exact $p$ Value * |
|---|---|---|---|---|---|---|---|
| | | Summer | Fall | Winter | Spring | | |
| Sleep Deprivation | | | | | | | |
| Total | Yes | 31 (21.5) | 47 (32.6) | 8 (5.6) | 58 (40.3) | 6.538 | 0.091 |
| | No | 105 (24.8) | 166 (39.2) | 30 (7.1) | 122 (28.8) | | |
| 18–39 years | Yes | 16 (22.5) | 16 (22.5) | 4 (5.6) | 35 (49.3) | 9.058 | 0.040 |
| | No | 65 (27.3) | 82 (34.5) | 19 (8.0) | 72 (30.3) | | |
| 40–49 years | Yes | 7 (24.1) | 12 (41.4) | 1 (3.4) | 9 (31.0) | 0.249 | 0.966 |
| | No | 13 (20.0) | 28 (43.1) | 3 (4.6) | 21 (32.3) | | |
| 50–59 years | Yes | 3 (13.0) | 11 (47.8) | 2 (8.7) | 7 (30.4) | 1.485 | 0.693 |
| | No | 18 (24.0) | 33 (44.0) | 7 (9.3) | 17 (22.7) | | |
| 60 years and older | Yes | 5 (23.8) | 8 (38.1) | 1 (4.8) | 7 (33.3) | 1.140 | 0.670 |
| | No | 9 (20.0) | 23 (51.1) | 1 (2.2) | 12 (26.7) | | |
| Male | Yes | 18 (23.7) | 18 (23.7) | 7 (9.2) | 33 (43.4) | 8.386 | 0.038 |
| | No | 45 (26.2) | 67 (39.0) | 14 (8.1) | 46 (26.7) | | |
| Female | Yes | 13 (19.1) | 29 (42.6) | 1 (1.5) | 25 (36.8) | 3.798 | 0.303 |
| | No | 60 (23.9) | 99 (39.4) | 16 (6.4) | 76 (30.3) | | |
| Insomnia | | | | | | | |
| Total | Yes | 25 (22.9) | 38 (34.9) | 4 (3.7) | 42 (38.5) | 3.145 | 0.409 |
| | No | 107 (23.4) | 175 (38.2) | 32 (7.0) | 144 (31.4) | | |
| 18–39 years | Yes | 14 (25.0) | 16 (28.6) | 3 (5.4) | 23 (41.1) | 1.307 | 0.799 |
| | No | 67 (26.3) | 82 (32.2) | 20 (7.8) | 86 (33.7) | | |
| 40–49 years | Yes | 7 (22.6) | 12 (38.7) | 1 (3.2) | 11 (35.5) | 0.488 | 0.941 |
| | No | 12 (19.7) | 27 (44.3) | 3 (4.9) | 19 (31.1) | | |
| 50–59 years | Yes | 1 (7.7) | 9 (69.2) | 0 (0.0) | 3 (23.1) | 4.455 | 0.311 |
| | No | 19 (22.4) | 35 (41.2) | 8 (9.4) | 23 (27.1) | | |
| 60 years and older | Yes | 3 (33.3) | 1 (11.1) | 0 (0.0) | 5 (55.6) | 6.321 | 0.054 |
| | No | 9 (15.8) | 31 (54.4) | 1 (1.8) | 16 (28.1) | | |

**Table 3.** *Cont.*

| Age Group/Sex | Sleep Problem Yes/No | Seasons *n* (%) | | | | X², df = 3 | Exact *p* Value * |
|---|---|---|---|---|---|---|---|
| | | Summer | Fall | Winter | Spring | | |
| Male | Yes | 11 (27.5) | 13 (32.5) | 2 (5.0) | 14 (35.0) | 0.912 | 0.868 |
| | No | 49 (23.7) | 73 (35.3) | 18 (8.7) | 67 (32.4) | | |
| Female | Yes | 14 (20.3) | 25 (36.2) | 2 (2.9) | 28 (40.6) | 2.850 | 0.478 |
| | No | 58 (23.1) | 102 (40.6) | 14 (5.6) | 77 (30.7) | | |
| | Excessive daytime sleepiness | | | | | | |
| Total | Yes | 17 (16.3) | 37 (35.6) | 8 (7.7) | 42 (40.4) | 5.089 | 0.148 |
| | No | 116 (24.7) | 177 (37.7) | 28 (6.0) | 148 (31.6) | | |
| 18–39 years | Yes | 8 (17.4) | 15 (32.6) | 5 (10.9) | 18 (39.1) | 2.831 | 0.376 |
| | No | 73 (27.3) | 83 (31.1) | 17 (6.4) | 94 (35.2) | | |
| 40–49 years | Yes | 3 (13.0) | 9 (39.1) | 1 (4.3) | 10 (43.5) | 2.330 | 0.494 |
| | No | 17 (23.9) | 31 (43.7) | 3 (4.2) | 20 (28.2) | | |
| 50–59 years | Yes | 2 (11.8) | 10 (58.8) | 1 (5.9) | 4 (23.5) | 2.000 | 0.655 |
| | No | 19 (23.2) | 34 (41.5) | 7 (8.5) | 22 (26.8) | | |
| 60 years and older | Yes | 4 (22.2) | 3 (16.7) | 1 (5.6) | 10 (55.6) | 9.901 | 0.007 |
| | No | 7 (14.3) | 29 (59.2) | 1 (2.0) | 12 (24.5) | | |
| Male | Yes | 7 (15.2) | 15 (32.6) | 4 (8.7) | 20 (43.5) | 3.810 | 0.259 |
| | No | 55 (26.6) | 71 (34.3) | 17 (8.2) | 64 (30.9) | | |
| Female | Yes | 10 (17.2) | 22 (37.9) | 4 (6.9) | 22 (37.9) | 2.088 | 0.528 |
| | No | 61 (23.3) | 106 (40.5) | 11 (4.2) | 84 (32.1) | | |

* *p* values calculated using R function fisher.test using 2000 simulations.

A significant variation in sleep problems was observed in the age and sex stratification (Table 3). Further analysis was conducted comparing two proportions using the Z test. There was a significantly high prevalence of sleep deprivation in the age group of 18–39 years ($p = 0.003$) and among males ($p = 0.009$) in the spring season. Males had significantly lower prevalence of sleep deprivation in the fall ($p = 0.019$). There was significantly low prevalence of insomnia among participants aged 60 years and older age group in the fall ($p = 0.016$), while excessive daytime sleepiness was significantly higher for this age group in the spring ($p = 0.016$) and significantly lower in the fall ($p = 0.002$).

Of the participants, 360 responded to the question related to how much their sleep changes with the seasons. Of those, 68.6% reported a slight, moderate, very, or extremely marked change, with 10.3% reporting an extremely marked change. About one-third (31.4%) of the participants reported that there was no change in their sleep patterns across seasons (Table 4). Sleep change by seasons was further analysed using a binary (very or extremely marked, or no-to-moderate change) logistic-regression model to determine associations with demographic, social, and environmental factors (Table 5).

**Table 4.** How much sleep changes with seasons (*n* = 360).

| Sleep Change | Frequency (%) |
|---|---|
| 0 (no change) | 113 (31.4) |
| 1 (slight) | 51 (14.2) |
| 2 (moderate) | 101 (28.1) |
| 3 (very) | 58 (16.1) |
| 4 (extremely marked change) | 37 (10.3) |

**Table 5.** Association among demographic, social, and environmental factors and sleep change with seasons (*n* = 360).

| Variable | Sleep Change with the Seasons | | | Unadjusted Odds Ratio (95% CI) | Adjusted Odds Ratio (95% CI) |
|---|---|---|---|---|---|
| | Total | Yes (Very or Extremely Marked Change) | No (No to Moderate Change) | | |
| **Age Groups, in Years** | | | | | |
| 18–29 | 117 (32.5) | 26 (27.4) | 91 (34.3) | 1.36 (0.56, 3.27) | 1.36 (0.55, 3.35) |
| 30–39 | 82 (22.8) | 27 (28.4) | 55 (20.8) | 2.33 (0.96, 5.68) | 2.40 (0.97, 5.96) |
| 40–49 | 58 (16.1) | 18 (18.9) | 40 (15.1) | 2.14 (0.83, 5.49) | 2.02 (0.76, 5.37) |
| 50–59 | 57 (15.8) | 16 (16.8) | 41 (15.5) | 1.85 (0.71, 4.82) | 1.65 (0.62, 4.37) |
| 60 years and older | 46 (12.8) | 8 (8.4) | 38 (14.3) | 1.00 | 1.00 |
| **Sex** | | | | | |
| Male | 158 (43.9) | 39 (43.9) | 119 (44.9) | 0.85 (0.53, 1.37) | 0.80 (0.49, 1.33) |
| Female | 202 (56.1) | 56 (58.9) | 146 (55.1) | 1.00 | 1.00 |
| **Smoking Status** | | | | | |
| Current smoker | 259 (72.5) | 66 (69.5) | 193 (73.7) | 0.93 (0.49, 1.80) | - |
| Ex-smoker | 42 (11.8) | 14 (14.7) | 28 (10.7) | 1.37 (0.57, 3.27) | - |
| Never smoker | 56 (15.7) | 15 (15.8) | 41 (15.6) | 1.00 | - |
| **Body-Mass Index** | | | | | |
| Obese | 151 (45.6) | 32 (37.6) | 119 (48.4) | 0.62 (0.34, 1.12) | - |
| Overweight | 91 (27.5) | 26 (30.6) | 65 (26.4) | 0.92 (0.48, 1.74) | - |
| Neither overweight or obese | 89 (26.9) | 27 (31.8) | 62 (25.2) | 1.00 | - |
| **Employment Status** | | | | | |
| Full-time, part-time, or self-employed | 103 (29.3) | 29 (31.9) | 74 (28.5) | 1.31 (0.64, 2.68) | - |
| Social assistance or unemployment insurance | 106 (30.2) | 27 (29.7) | 79 (30.4) | 1.14 (0.55, 2.35) | - |
| Unemployed | 77 (21.9) | 20 (22.0) | 57 (21.9) | 1.17 (0.54, 2.52) | - |
| Other, including retired or home maker | 65 (18.5) | 15 (16.5) | 50 (19.2) | 1.00 | - |
| **Educational Attainment** | | | | | |
| Less than secondary-school graduation | 141 (39.8) | 34 (36.6) | 108 (40.9) | 0.71 (0.40, 1.24) | - |
| Secondary-school graduation | 108 (30.3) | 26 (28.0) | 82 (31.1) | 0.71 (0.39, 1.30) | - |
| Some university, completed university degree, completed technical school | 107 (30.0) | 33 (35.5) | 74 (28.0) | 1.00 | - |
| **Money Left at the End of the Month** | | | | | |
| Some money | 74 (20.8) | 16 (17.0) | 58 (22.1) | 0.61 (0.32, 1.14) | 0.71 (0.37, 1.38) |
| Just enough money | 77 (21.6) | 14 (14.9) | 63 (24.0) | 0.49 (0.26, 0.94) | 0.50 (0.25, 0.97) |
| Not enough money | 205 (57.6) | 64 (68.1) | 141 (53.8) | 1.00 | 1.00 |
| **Damage Caused by Dampness** | | | | | |
| Yes | 213 (59.7) | 67 (71.3) | 146 (55.5) | 1.99 (1.20, 3.31) | 1.84 (1.08, 3.14) |
| No | 144 (40.3) | 27 (28.7) | 117 (44.5) | 1.00 | 1.00 |
| **Mouldy or Musty Smell in House** | | | | | |
| Yes | 191 (53.4) | 59 (62.8) | 132 (50.0) | 1.69 (1.04, 2.73) | - |
| No | 167 (46.6) | 35 (37.2) | 132 (50.0) | 1.00 | - |
| **Signs of Mould in House** | | | | | |
| Yes | 184 (51.3) | 59 (63.4) | 125 (47.3) | 1.93 (1.19, 3.14) | - |
| No | 173 (48.5) | 34 (36.6) | 139 (52.7) | 1.00 | - |
| **Smoke Inside Home** | | | | | |
| Yes | 151 (42.3) | 38 (40.4) | 113 (43.0) | 0.90 (0.56, 1.45) | - |
| No | 206 (57.7) | 56 (59.6) | 150 (57.0) | 1.00 | - |

**Table 5.** *Cont.*

| Variable | Sleep Change with the Seasons | | | Unadjusted Odds Ratio (95% CI) | Adjusted Odds Ratio (95% CI) |
| | Total | Yes (Very or Extremely Marked Change) | No (No to Moderate Change) | | |
|---|---|---|---|---|---|
| **Crowding Index** | | | | | |
| >1 person/bedroom | 266 (75.6) | 68 (73.1) | 198 (76.4) | 0.84 (0.49, 1.44) | - |
| ≤1 person/bedroom | 86 (24.4) | 25 (26.9) | 61 (23.6) | 1.00 | - |
| **Have Access to Off-Reserve Doctor** | | | | | |
| Yes | 296 (85.3) | 83 (89.2) | 213 (83.9) | 1.60 (0.77, 3.34) | - |
| No | 51 (14.7) | 10 (10.8) | 41 (16.1) | 1.00 | - |
| **Have Access to Doctor at On-Reserve Clinic** | | | | | |
| Yes | 273 (80.5) | 78 (86.7) | 195 (78.3) | 1.80 (0.91, 3.55) | - |
| No | 66 (19.5) | 12 (13.3) | 54 (21.7) | 1.00 | - |
| **Have Access to Nurse** | | | | | |
| Yes | 261 (80.8) | 74 (85.1) | 187 (79.2) | 1.49 (0.76, 2.91) | - |
| No | 62 (19.2) | 13 (14.9) | 49 (20.8) | 1.00 | - |
| **Difficulty Gaining Access to Medical Specialist** | | | | | |
| Yes | 84 (23.5) | 28 (29.8) | 56 (21.3) | 1.57 (0.92, 2.67) | - |
| No | 273 (76.5) | 66 (70.2) | 207 (78.7) | 1.00 | - |

The bi- and multivariate-regression model results are presented in Table 5. Compared to the older age group, younger age groups were more likely to report seasonal changes in sleep, but they were not statistically significant. In addition, participants who reported having just enough money left at the end of the month were significantly less likely to report seasonal changes in sleep compared to those who reported not having enough money left at the end of the month. Dampness-related damage to the house was also a significant factor for change in sleep patterns across seasons (Table 5).

## 4. Discussion

In this study, participants reported sleeping the least during the months of May–August, and sleeping the most during the months of November–February. This was further confirmed by sleep hours and the lower proportion of recommended hours of sleep during the spring and summer, and a higher proportion of longer duration of sleep during the fall and winter months. There was no reported significant variation of sleeping onset and wake-up times by season. Reported hours of sleep by seasons confirmed that there was a significant difference in sleep hours during spring and summer. Overall, there were no significant differences in the prevalence of sleep deprivation, insomnia, and EDS by season. When stratified by age group and sex, some differences existed in the prevalence of sleep problems by season. More than two-thirds of participants reported that there was a change in sleep patterns across seasons, and about 10% reported an extremely marked change. Changes in sleep patterns by season were related to age, money left at the end of the month, damage caused by dampness in house, and having access to a nurse.

Seasonal changes in sleep problems were studied in many countries [5,14,34–37], and studies reported an effect of seasonality on sleep quality. Putilov reported seasonal-variation effects on sleep problems such as daytime sleepiness, difficulties falling asleep, difficulty staying asleep, and premature awakenings [37]. Another author reported that self-reported measures indicated moderately to strong seasonal differences in insomnia and fatigue prevalence, but no seasonal changes were observed in sleep duration or night awakenings [5]. Few studies reported a strong effect of seasonality on poorer sleep during the winter months [34,35,38]. Pallesen et al. [35] demonstrated that sleep-onset problems and insomnia were more frequent in December compared with June. Husby and Lingjaerde

found an increase in the prevalence of insomnia during winter, but also more sleeplessness during summer [34]. Lukmanjini et al. [14] reported that seasonal variation for insomnia symptoms (trouble falling or staying asleep, or sleeping too much) were similar among age groups of 12–24 and 25+ years. However, the younger age group reported a higher proportion of insomnia symptoms compared with the older group. A high proportion of sleep problems was also observed among adults during the fall and winter months [14]. Seasonal effects on sleep problems such as difficulty initiating sleep and excessive daytime sleepiness, in young (15–39 years) and middle (40–64 years) age groups were reported by Suzuki et al. [4]. A significant seasonal difference was found for the prevalence of insomnia symptoms [4]. Reported insomnia symptoms were more prevalent in spring than in fall and winter. The prevalence of insomnia was also higher in summer than it was in fall and winter. In contrast to these findings, those presented by Sivertsen et al. [39] reported no evidence of seasonal variation on reporting insomnia systems or time in bed in a geographic region of Norway with large seasonal differences in daytime light. In this study, there were no significant differences in the prevalence of sleep deprivation, insomnia, and EDS by season. When stratified by age group and sex, some differences existed in prevalence of sleep problems by season, which need to be further explored.

Several studies reported that there were seasonal changes in sleep onset and offset [5,40–44]. According to Hashizaki et al., sleep-onset time did not show clear seasonal variation, but sleep offset time showed a seasonal change in winter. In addition, sleep-offset time correlated well with sunrise time [40]. Seasonal variation in sleep quality (wake time after sleep onset (WASO) and sleep efficiency (SE, the ratio of total sleep time to time spent in bed × 100)) were also observed. The day with maximal WASO during summer, meaning the worst sleep quality, was nearly synchronous with the highest day temperature, whereas minimal SE also corresponded to the highest day temperature. A similar pattern was observed in the winter corresponding to the coldest temperature days for WASO and SE [40]. Friborg et al. [5] reported that there was low to moderately strong seasonal changes in the time to rise and bedtime, sleep efficiency, and sleep-onset latency in the northern latitude (Norway 69° 39′ N). Studies showed that cold and hot temperatures impact human sleep [43,44]. Other studies showed a later wake-up time and longer sleep duration in winter compared to summer [41,42]. One study reported delayed circadian rhythms due to reduced exposure to morning light during fall and winter [45]. Yetish et al. [6] reported that morning ambient temperature affected the sleep offset time independent from sunlight. Daily variation in sleep duration was strongly linked to the time of sleep onset. In contrast to these studies, this study did not show a significant variation in sleep onset and wake-up times by season.

Numerous studies reported seasonal variation in sleep duration [1,4,46]. Hardin et al. [1] reported that there was seasonal variation in sleep among the general population of the United States, and that people sleep more in the winter. Seasonal changes in sleep duration were reported by Suzuki et al. [4] in a Japanese population, with longer sleep duration in winter and shorter sleep duration in summer. Men slept significantly longer than women did, and slept significantly less in summer than in fall and winter. On the other hand, women slept significantly longer in winter than spring, summer, and fall [4]. Seasonal variations in sleep time and sleep duration that were reported in other traditional societies [46] were approximately 7 h of sleep in summer and 8.5 h of sleep in winter. Similar to these studies, in this present study, participants reported sleeping the least during the spring and summer months, and sleeping the most during fall and winter. This was further confirmed by hours of sleep and the lower proportion of recommended hours of sleep during the spring and summer, and a higher proportion of longer duration of sleep during the fall and winter months.

In this study, 26.4% reported a very or extremely marked change in sleep patterns across the seasons, and about 10% reported an extremely marked change. There were no significant associations between age group and changes in sleep patterns by season. In contrast, Suzuki et al. [4] reported seasonality changes in sleep duration, which were

influenced by age, with those in the middle (40–64 years) and older age (65–89 years) groups affected, but not those in the younger age (15–39 years) group. Sleep duration in the older age group was longer than that in the middle and younger age groups for every season. The middle-aged group slept longer in winter than in spring and summer, and more in fall than in summer. There were no significant differences in sleep duration among seasons in the younger age group. Suzuki et al. identified sex differences for seasonal variation in sleep duration [4]. In this study, change in sleep patterns was not influenced by sex.

Some authors reported an association between socioeconomic status and sleep issues [47,48]. Lallukka et al. [47] reported that a disadvantaged socioeconomic position among adults, in particular income and employment status, was associated with poorer sleep. Another study reported that a lower socioeconomic status was associated with higher rates of sleep complaints [48]. However, there were no previous investigations available with seasonal changes in sleep patterns and socioeconomic status. This study identified that socioeconomic indices such as those with 'just enough' money left at the end of the month were less likely to change their sleep patterns by season compared to 'not enough' money left at the end of the month. In First Nation communities, job insecurity and financial insecurity are common, and this can negatively impact mental health and wellbeing, contributing to poorer health behaviours, including sleep disruption [49,50]. Another study also indicated that Indigenous men are less likely to have permanent employment and more likely to hold a seasonal job than non-Indigenous adults are [51]. Moreover, this study observed that damage caused by dampness in the house was more likely to be associated with changes in sleep patterns by season. Dampness is caused by excess moisture. The most common form of dampness is condensation, which can form when warm moist air touches a cold internal wall or surface. Dampness can form quickly in winter weather, especially if there is an issue with the exterior of the building. Studies showed that dampness and mould worsen sleep problems among adults [52] and children [53]. One possible explanation given by Wang at al. [52] was that a damp and mouldy environment can cause irritation and inflammation in mucus membranes or airways, which is a risk factor for sleep disturbances. Therefore, important socioeconomic indices, financial insecurity, seasonal employment, and housing-environment conditions warrant further investigation to confirm the reasons for changes in sleep patterns by season.

However, this study did not observe change in sleep patterns with participants who had access to any healthcare providers. Nurses are often the communities' main point of contact with the healthcare system [54]. In Saskatchewan, there are primary-care physicians that go to the communities but not every day; specialist care frequently requires First Nations peoples to travel to Regional Centres or Tertiary Care Centres.

*Limitations*

Data on sleep onset and wakeup times, sleep duration, seasonal variation, and sleep problems were based on self-reporting, which could have biased the findings. Sleep duration was not assessed separately for work or school and free days, or for weekends or holidays. The difference between seasons in the number of working days may have confounded the results. Many other factors, including psychiatric diagnoses, physical comorbidities, the use of sedatives, and other lifestyle factors, were not assessed. We do not have repeated data for participants for all four seasons. Therefore, we were not able to explore the individual changes by season, but could only identify population-level changes by seasons. Due to the small sample size of changes by season outcome, we were unable to see whether there were group differences according to when people sleep longer or shorter, and whether the same factors of changes by season hold in different sleep-duration groups.

## 5. Conclusions

This study reported seasonal changes in sleep patterns in two Saskatchewan First Nation communities. These changes in sleep patterns were influenced by age, income

status, dampness in house, and access to a nurse. This information is helpful for practice improvement by healthcare providers in treating and managing patients with sleep disorders, policy development, and future research priorities in and with these communities.

**Author Contributions:** Conceptualization, J.A.D., S.A., M.K., P.P., D.C.R., S.K., N.K., M.F., R.S., and the First Nations Sleep Health Project Team; data curation, B.P.R. and K.M.; formal analysis, C.P.K.; funding acquisition, J.A.D., S.A., M.K., and P.P.; investigation, J.A.D., S.A., M.F., M.K., and P.P.; methodology, J.A.D., P.P., S.A., M.K., C.P.K., and M.F.; project administration, P.P.; resources, J.S., C.B., R.S., M.F., and T.S.-W.; supervision, J.A.D. and P.P.; visualization, S.A., J.S., C.B., and V.R.R.; writing—original draft, C.P.K.; writing—review and editing, J.A.D., C.P.K., B.P.R., K.M., S.A., D.C.R., S.K., N.K., J.S., C.B., V.R.R., M.F., M.K., T.S.-W., and P.P. All authors have read and agreed to the published version of the manuscript.

**Funding:** This research was funded by grant from the Canadian Institutes of Health Research "Assess, Redress, Reassess: Addressing Disparities in Sleep Health among First Nations People", CIHR MOP-391913-IRH-CCAA-11829-FRN PJT-156149.

**Institutional Review Board Statement:** The study was conducted according to the guidelines of the Declaration of Helsinki, and approved by the Biomedical Research Ethics Board of University of Saskatchewan (Bio #18-110; date of approval, 21 June 2018).

**Informed Consent Statement:** Written informed consent has been obtained from all participants involved in the study.

**Data Availability Statement:** The summarized data presented in this study are available on request from the corresponding author. The data are not publicly available due to the agreement with two participating communities.

**Acknowledgments:** The First Nations Sleep Health Project Team consists of: James A Dosman, (designated principal investigator, University of Saskatchewan, Saskatoon, SK, Canada); Punam Pahwa, (coprincipal investigator, University of Saskatchewan, Saskatoon SK, Canada); Malcolm King, (coprincipal investigator, University of Saskatchewan, Saskatoon, SK, Canada), Sylvia Abonyi, (coprincipal investigator, University of Saskatchewan, Saskatoon, SK, Canada); coinvestigators: Mark Fenton, Chandima P Karunanayake, Shelley Kirychuk, Niels Koehncke, Joshua Lawson, Robert Skomro, Donna C Rennie, Darryl Adamko. Collaborators: Roland Dyck, Thomas Smith-Windsor, Kathleen McMullin, Rachana Bodani, John Gjerve, Bonnie Janzen, and Vivian R Ramsden, Gregory Marchildon, and Kevin Colleaux. Project manager: Brooke P Russell. Community partners: Jeremy Seeseequasis, Clifford Bird, Roy Petit, Edward Henderson, Raina Henderson, Dinesh Khadka. We are grateful for the contributions from elders and the community leaders who facilitated the engagement necessary for the study, and all participants who engaged in this study.

**Conflicts of Interest:** The authors declare no conflict of interest. The funders had no role in the design of the study; in the collection, analyses, or interpretation of data; in the writing of the manuscript, or in the decision to publish the results.

## Abbreviations

| | |
|---|---|
| N | North |
| W | West |
| FNSHP | First Nations Sleep Health Project |
| SPAQ | Seasonal Pattern Assessment Questionnaire |
| EDS | Excessive daytime sleepiness |
| ISI | Insomnia Severity Index |
| ESS | Epworth Sleepiness Scale |
| SD | Standard Deviation |
| CI | Confidence Interval |
| WASO | Wake time after sleep onset |
| SE | Sleep efficiency |

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
