# Peer review of "Seasonal Changes in Sleep Patterns in Two Saskatchewan First Nation Communities"

_2624-5175, doi:10.3390/clockssleep3030029_

Round 1

Reviewer 1 Report

Manuscript ID: clockssleep-1292687

Title: Seasonal changes in sleep patterns in two Saskatchewan First Nation communities

Journal: Clocks & Sleep

  1. Authors’ reply to my previous comment number 1 on the Introduction.

It seems that the expected results are not reported at lines 65-68.

  1. Authors’ reply to my previous comment number 1 on the Methods.

It seems that Authors did not correctly compute the midpoint of sleep as reported at line 145. In order to properly calculate such parameter, they should take a look at the Supplemental Data of the study by Roenneberg and colleagues (Current Biology, 2004, Volume 14, Issue 24, Pages R1038-R1039).

Author Response

Reviewer 1 comments:

Open Review

English language and style

( ) Extensive editing of English language and style required  
( ) Moderate English changes required  
( ) English language and style are fine/minor spell check required  
(x) I don't feel qualified to judge about the English language and style  

Comments and Suggestions for Authors

Manuscript ID: clockssleep-1292687

Title: Seasonal changes in sleep patterns in two Saskatchewan First Nation communities

Journal: Clocks & Sleep 

  1. Authors’ reply to my previous comment number 1 on the Introduction.

It seems that the expected results are not reported at lines 65-68.

Response: As reviewer suggested following statements were added to the Introduction to indicate the expected results, but results are presented in the “Results” section and discussed in “Discussion” section. However, the following statement is changed. See lines 74-77.

“This paper examines the seasonal effects on sleep time, sleep duration and sleep problems in two Cree First Nation communities in Saskatchewan, Canada and identifies the association between seasonal changes in sleep patterns and demographic, social and environmental factors.”

“This paper examines the seasonal effects on sleep time, sleep duration and sleep problems in two Cree First Nation communities in Saskatchewan, Canada and identifies the association between seasonal changes in sleep patterns and demographic, social and environmental factors. The expected outcomes of this paper will help to identify the seasonal effects on sleep and the factors related to seasonal changes in sleep patterns.”

  1. Authors’ reply to my previous comment number 1 on the Methods.

It seems that Authors did not correctly compute the midpoint of sleep as reported at line 145. In order to properly calculate such parameter, they should take a look at the Supplemental Data of the study by Roenneberg and colleagues (Current Biology, 2004, Volume 14, Issue 24, Pages R1038-R1039).

Response: According to Roenneberg and colleagues (Current Biology, 2004, Volume 14, Issue 24, Pages R1038-R1039) the definition is midpoint is given below.

“Definitions: The mid-sleep time on free days (MSF) is calculated from questions concerning sleep onset and awakening time on days on which there are no work or social obligations. MSF is the mid-point between these two times.”

Unfortunately we did not collect information on sleep onset and awakening time on days which there is no work or social obligations. We asked three questions about during the past month, usual bedtime at night, usual time of wake up in the morning and time to fall asleep at night. We reported median sleep duration from the available data.

Reviewer 2 Report

The authors of this manuscript reported seasonal changes in sleep patterns in two Saskatchewan communities, and they analyzed the possible contribution of different factors in sleep seasonality and sleep problems. They found a significant difference in sleep duration between Spring and Summer, but did not find the seasonal differences in sleep timing, excessive daytime sleepiness, etc. The results also pointed at the association of sleep problems with socioeconomic indexes, financial insecurity, seasonal employment, housing conditions and other socio-psychological rather than biological and geophysical factors. I think the authors adequately responded to the flaws of their initial version of this manuscript noted by other reviewers, and now the manuscript can be published in the journal in its present (revised) form.

Reviewer 3 Report

This is perfectly constructed and presented study on  seasonal changes in sleeping time, adding important knowledge to state-of-the art . Minor editorial  remark :v. 106: there is one excessive  point (after „ ?”)

Author Response

Reviewer 2 Comments

Open Review

English language and style

( ) Extensive editing of English language and style required  
( ) Moderate English changes required  
(x) English language and style are fine/minor spell check required  
( ) I don't feel qualified to judge about the English language and style  

Yes

Can be improved

Must be improved

Not applicable

Is the content succinctly described and contextualized with respect to previous and present theoretical background and empirical research (if applicable) on the topic?

(x)

( )

( )

( )

Are the research design, questions, hypotheses and methods clearly stated?

(x)

( )

( )

( )

Are the arguments and discussion of findings coherent, balanced and compelling?

(x)

( )

( )

( )

For empirical research, are the results clearly presented?

(x)

( )

( )

( )

Is the article adequately referenced?

(x)

( )

( )

( )

Are the conclusions thoroughly supported by the results presented in the article or referenced in secondary literature?

( )

( )

( )

( )

Comments and Suggestions for Authors

This is perfectly constructed and presented study on  seasonal changes in sleeping time, adding important knowledge to state-of-the art . Minor editorial  remark :v. 106: there is one excessive  point (after „ ?”)

Response: Missing “ was added. Please see line 117 in page 3.

This manuscript is a resubmission of an earlier submission. The following is a list of the peer review reports and author responses from that submission.

Round 1

Reviewer 1 Report

This interesting and well-written manuscript uses an important survey database from two Cree First Nation communities in Saskatchewan, Canada. The results are interesting but I think could be improved by considering some further issues and questions. 

It would be interesting to know when/what season respondents completed the survey, for a number of reasons, including if answers varied by when survey was done (although I note no difference in bed times across seasons, which in itself raises questions about how much change there actually is across seasons)

Although more people report sleeping less in summer and more in winter, about 20-25% report the opposite. Are these people different? It would be interesting to look at whether there are group differences according to when people sleep longer/shorter. This may be particularly important when constructing the regression models, as the predictors maybe different for different groups of “change” according to the season of most change. Given that socio-economic factors were related to more sleep change (house damage by dampness and “not enough money” increased odds of seasonal sleep change) it might be important to see if those factors hold in different groups – e.g. “long winter/short summer”; short winter/long summer”

This could also be of relevance for the potential translational aspects of the study mentioned in the discussion, i.e. developing policies and future research, which may be different for people depending on the season.

Was the choice to construct the binary outcome for the regression model splitting at any/no change pre-specified, and/or is there a justification for doing so? I would think there are arguments to say a “slight” change is probably insignificant and is qualitatively (and maybe physiologically) different to both extreme and extremely marked (categories 3 and 4). Was consideration given to a different binary outcome split or an ordinal regression model (appreciating numbers may be an issue).

In the Discussion, the authors speculate that “….. this study observed higher odds of change in sleep patterns with participants who had access to the nurse, which we can speculate being because the nurse will help to detect and direct the participants to treatment for their sleep problems”. It is not clear to me why the nurse detecting sleep problems will be associated with more change in sleep seasonally, as compared to lesschange due to, for example, advice to continue regular sleep habits despite changes in sunlight, etc. It may be worth some comment on the role of community nurses in First Nations’ healthcare in Saskatchewan for those not familiar with the Canadian healthcare system. Is this variable a proxy marker of socio-economic status of the individual or the community?

Reviewer 2 Report

Manuscript ID: clockssleep-1214976

Title: Seasonal changes in sleep patterns in two Saskatchewan First Nation communities

Journal: Clocks & Sleep

Abstract

  1. Line 27. Authors should add the number of males and females as well as the age range.

Introduction

  1. Line 66. Authors should add the expected results.

Methods

  1. Lines 101-103. Authors should compute the midpoint of sleep as marker of the sleep timing.

Results

  1. Lines 130-131. Authors should clarify that their study is a secondary analysis of previously collected data, starting from the title.

  1. Lines 132-133. The age range is wide. Based on age distribution, authors should only focus on age groups larger in size.

  1. Since Table 1 and Figure 2 report the same data, authors should delete Table 1.

  1. Line 153. Looking at Table 2, it seems that most participants reported their usual bedtime between midnight and 2 AM.

  1. Table 4. Authors should adopt the same age groping with reference to each dependent variable.

  1. Table 5. In order help the reader, authors should add the meaning of values 1, 2 and 3.

  1. Lines 179-180. I am wondering how authors computed the percentage value of 68.6%.

  1. Table 6. I am wondering why authors did not consider the same independent variables (e.g., body mass index and employment status) with reference to each dependent variable.

  1. Lines 189-196. I am wondering which could be the meaning of these results.

Discussion

  1. Within this section, authors should explain the observed results.

  1. Lines 209-210. Authors should explain the reasons why changes in sleep pattern by season were related to those factors.